# Independence of a Marine Unicellular Diazotroph to the Presence of NO_3_^−^

**DOI:** 10.3390/microorganisms9102073

**Published:** 2021-10-01

**Authors:** Sophie Rabouille, Benjamin Randall, Amélie Talec, Patrick Raimbault, Thierry Blasco, Amel Latifi, Andreas Oschlies

**Affiliations:** 1Laboratoire d’Océanographie Microbienne (LOMIC), CNRS, Sorbonne Université, F-66650 Banyuls-sur-Mer, France; 2Laboratoire d’Océanographie de Villefranche (LOV), CNRS, Sorbonne Université, F-06230 Villefranche-sur-Mer, France; ben.randall1992@gmail.com (B.R.); amelie.talec@imev-mer.fr (A.T.); thierry.blasco@upmc.fr (T.B.); 3Aix Marseille Université, CNRS/INSU, Université de Toulon, IRD, Mediterranean Institute of Oceanography (MIO) UM 110, 13288 Marseille, France; patrick.raimbault@mio.osupytheas.fr; 4Laboratoire de Chimie Bactérienne (LCB), Aix Marseille Université, CNRS, 13284 Marseille, France; latifi@imm.cnrs.fr; 5GEOMAR Helmholtz Centre for Ocean Research Kiel, 24105 Kiel, Germany; aoschlies@geomar.de

**Keywords:** nitrogen fixation, marine cyanobacteria, nitrate uptake, *Crocosphaera*

## Abstract

Marine nitrogen (N_2_) fixation was historically considered to be absent or reduced in nitrate (NO_3_^−^) rich environments. This is commonly attributed to the lower energetic cost of NO_3_^−^ uptake compared to diazotrophy in oxic environments. This paradigm often contributes to making inferences about diazotroph distribution and activity in the ocean, and is also often used in biogeochemical ocean models. To assess the general validity of this paradigm beyond the traditionally used model organism *Trichodesmium* spp., we grew cultures of the unicellular cyanobacterium *Crocosphaera watsonii* WH8501 long term in medium containing replete concentrations of NO_3_^−^. NO_3_^−^ uptake was measured in comparison to N_2_ fixation to assess the cultures’ nitrogen source preferences. We further measured culture growth rate, cell stoichiometry, and carbon fixation rate to determine if the presence of NO_3_^−^ had any effect on cell metabolism. We found that uptake of NO_3_^−^ by this strain of *Crocosphaera* was minimal in comparison to other N sources (~2–4% of total uptake). Furthermore, availability of NO_3_^−^ did not statistically alter N_2_ fixation rate nor any aspect of cell physiology or metabolism measured (cellular growth rate, cell stoichiometry, cell size, nitrogen fixation rate, nitrogenase activity) in comparison to a NO_3_^−^ free control culture. These results demonstrate the capability of a marine diazotroph to fix nitrogen and grow independently of NO_3_^−^. This lack of sensitivity of diazotrophy to NO_3_^−^ suggests that assumptions often made about, and model formulations of, N_2_ fixation should be reconsidered.

## 1. Introduction

One of the key parameters classically considered to constrain the distribution of diazotrophs in the ocean is the availability, or lack thereof, of dissolved inorganic nitrogen (DIN). Two main arguments for this are brought forward in the literature. The primary reasoning is that the energetic demand of the uptake of DIN, such as ammonium (NH_4_^+^) and nitrate (NO_3_^−^) has been shown in oxic waters to be less costly than that of nitrogenase activity to convert N_2_ to NH_4_^+^ [1,2,3], and the diel synthesis of the nitrogenase complex imposes additional costs upon diazotrophic cells [2]. Logically, one might therefore assume that a diazotroph would, in oxic environments, always choose DIN over N_2_ whenever available. However, marine diazotrophs tend to be slow-growing compared to other phytoplankton, and are not the most efficient in taking up inorganic nutrients [4]. Therefore, when in the presence of non-limiting concentrations of nitrate, diazotrophs are likely to be outcompeted for DIN sources, and, following classical bottom-up reasoning, assumed unlikely to be found in nutrient rich areas of the ocean. This first argument, though, also applies to photo-autotrophic diazotrophs, and has lost genericity as photo-heterotrophic and heterotrophic diazotrophs have since been described [5,6,7,8], whose distribution can expand to NO_3_^−^-rich areas of the ocean [9].

The second line of thinking arose from seminal works on the easily observable, widely distributed diazotrophs such as *Trichodesmium* spp., which were the first to be maintained in culture and long constituted a reference for the N_2_ fixation process in situ [10]. Reduction of N_2_ fixation in the presence of DIN has been found in *Trichodesmium* spp. and is relatively well documented. A review [11] summarizing multiple culture and field experiments highlights that a substantial reduction in diazotrophy in *Trichodesmium* spp. occurred. The impact of DIN on nitrogenase activity was, however, infrequent and incomplete [12,13,14,15,16] in what we define here as being short term exposure, that is shorter than, or equivalent to, the time scale of the cell cycle. Although the term ‘inhibition’ has been used in the literature, it is unclear whether there is a real steric inhibition, sensu stricto, of the nitrogenase enzyme and how this inhibition would operate. We therefore refer to the more accurate term of ‘reduction’. This impact was also sensitive to the species of DIN. NH_4_^+^ was much more likely to induce a substantial reduction over short time scales, although NH_4_^+^ is far less prevalent than NO_3_^−^ in the ocean. NO_3_^−^ was less likely to cause high rates of reduction over these timescales. Over multiple generations of exposure to DIN, *Trichodesmium* tended to switch to DIN utilization over N_2_ fixation for the majority of its N acquisition needs (e.g.,[12,14,17]). An efficient switch to nitrate use is also reported in the unicellular strain *Cyanothece* sp. BG 043511, indicative of a reactive and plastic metabolism well adapted to the highly fluctuating conditions typical of coastal environments [3,4].

Both these lines of argument have often led to the conclusion that diazotrophs would eventually stop fixing N_2_ in the presence of DIN. It is then sensible to limit the process of N_2_ fixation to areas where NO_3_^−^ is not present in significant concentrations, but the above mentioned studies do not necessarily justify exclusion of diazotrophic organisms ecologically (that is, the organisms themselves) from such regions. Despite this uncertainty, this literature has often been used to constrain placement of diazotrophs in ecological and biogeochemical models of the ocean [18,19]. As a result, it is sometimes difficult to predict unrestrained N_2_ fixation [20] in areas where there are significant concentrations of NO_3_^−^, despite accumulated evidence from in situ observations [21]. As model placement of diazotrophy does not always align with marine observations, it is becoming clearer that this assumption of DIN controls on N_2_ fixation may not be entirely correct.

Paradigms of diazotrophy defined from observations of species such as *Trichodesmium* are currently coming under question due to recent expansion of the known phylogenetic pool of nitrogenase-producing, unicellular cyanobacteria [22,23,24]. Despite the need to test the consistency of conceptual and numerical model assumptions with diazotrophs newer to science, replication of this kind of study is currently impossible for the still uncultured diazotrophic cyanobacteria such as UCYN-A. A few culturable, marine unicellular cyanobacteria exist, however. One such is the UCYN-B *Crocosphaera watsonii*, naturally found in the open ocean. A small number of DIN limitation experiments have been performed on *C. watsonii* that show variability in results. Some experiments mirror, albeit to a seemingly lesser degree, the results obtained with *Trichodesmium* spp. experiments ([17] in long term exposure). Others suggest the ability to assimilate NO_3_^−^ and NH_4_^+^ efficiently whilst continuing to fix nitrogen with only a small reduction in N_2_ fixation ([25]; short term exposure) or reduction only contingent on low light conditions ([26]; long term exposure at 25 μmol quanta m^−2^ s^−1^). This variability in results actually points to a general uncertainty in our understanding of the effect of NO_3_^−^ exposure on *C. watsonii* and it is especially unclear how important NO_3_^−^ is to supporting the growth of *C. watsonii*.

We therefore designed a number of experiments with the aim of expanding our knowledge of the relationship between NO_3_^−^, N_2_ fixation, growth efficiency, and nitrogen metabolism. We grew *C. watsonii* in semi-continuous mode under different light regimes and with several concentrations of NO_3_^−^ (90 µM to 1000 µM) to test for any threshold level, which we compared to cultures grown under obligate diazotrophy, taken as reference.

## 2. Materials and Methods

All culture experiments were conducted on monocultures of *Crocosphaera watsonii* WH8501 [27,28], in acid-cleaned boro-silicate culture vessels, maintained under tightly controlled conditions of light and temperature in a Sanyo MLR 351 incubator as described in [29] and using the same experimental setup. Irradiance was provided by fluorescent tubes (Sanyo FL40SS W/37, Osaka, Japan). Each culture replicate received light from opposite faces and the cultures were placed in such a way that no shading effect between flasks occurred during the experiments. The original YBCII culture medium proposed by Chen et al. [30] was slightly modified so as not to contain any source of nitrogen (Fe-NH_4_-citrate was replaced with Fe-citrate). In each experiment described below, control (obligate diazotrophy, thereafter noted as NO_3_^−^-free) and treatment (NO_3_^−^-rich culture) cultures were initiated using the same culture medium, made fresh at the beginning of each experiment and as described in [31], to prevent any bias that might result from differences in medium preparation. Salinity was about 35 PSU and pH was ~8.2. Nitrate was thereafter amended in the NO_3_^−^-rich cultures. When dilutions were applied to refresh the cultures, the same medium devoid of NO_3_^−^ was used for all treatments. An appropriate addition of NO_3_^−^ was then made to each NO_3_^−^-rich culture individually to bring the NO_3_^−^ level back to that observed just before the dilution. Cultures were diluted at sufficient frequency (every 10 to 20 days in the present conditions) to maintain populations at exponential growth rate and at equivalent low densities, to limit any bias that would occur if cultures experienced differing irradiance, and also to prevent biases in the results that would actually be due to self-shading within flasks or to nutrient depletion. Cultures were grown long term in the presence of NO_3_^−^ for more than 10 generations in order to analyze the response of cells, not only upon exposure but also once they are acclimated to high concentrations of NO_3_^−^. This acclimation phase is particularly important to allow enough time for eventual genetic regulations to take place, and not to describe a metabolic response that would be a remnant of the dynamics of a previous conditions of growth. After acclimation, NO_3_^−^ uptake, N_2_ fixation rate, and a number of other physiological and metabolic parameters (cellular growth rate, cell size, N content, N fixation rate) were measured to test the effect of NO_3_^−^ on cell growth and activity. Cultures were inoculated at about 1 to 3·10^5^ cells. mL^−1^.

Population dynamics and average cell size in each population were monitored using a Multisizer 3 Coulter Counter (Beckman Coulter, Brea, CA, USA). A few samples were taken in the course of the experiment to control that bacterial contamination remained low, as in [31]. Elemental stoichiometry of organic carbon (POC) and nitrogen (PON) in the cultures were determined using a CHN analyzer (Perkin Elmer, Waltham, MA, USA). Samples (5.8 mL) were filtered onto pre-combusted (6 h at 450 °C) GF/C filters (Whatman, Buckinghamshire, UK), then dried and stored at 60 °C before analysis. At time points when POC and PON data were not available, cellular nitrogen (fmol-N cell^−1^) and carbon (fmol-C cell^−1^) contents were derived using the cell abundance measured at the time of sampling and the average C and N contents estimated with available data. The concentrations of nitrate (NO_3_^−^, μmol L^−1^) and nitrite (NO_2_, μmol L^−1^) in the cultures were monitored on a daily basis using a Technicon Auto-Analyzer calibrated on a range of [0–100] μmol NO_3_^−^ L^−1^. An automated dilution loop operated whenever the nitrate levels in the culture were beyond the calibration range, using the long tried-and-tested, automated data-acquisition system developed by Malara and Sciandra [32]. Nitrogenase activity was monitored using an on-line assay using the same setup as described in [31]. Culture samples of about 45 mL were filtered through a GF/F filter and placed in the cell incubator at the onset of the dark phase. The incubation temperature was maintained at 27 °C using a water bath and the sample experienced the same light cycle as in the growth incubator. Growth rates were estimated from exponential regressions on the changes in abundance data with time, with R^2^ the coefficient of determination of the exponential fit. Predictions of NO_3_^−^ consumption were derived from the observation of the population dynamics and the total, bulk N content in the population biomass. Under the hypothesis that N in the biomass comes from NO_3_^−^ uptake, the NO_3_^−^ uptake rate is predicted from the rate of biomass increase, by subtracting the amount of N incorporated in the biomass from the NO_3_^−^ pool, at each time step. This predicted curve of NO_3_^−^ disappearance from the medium simulates the consumption that would be necessary to support the nitrogen buildup in the biomass, should NO_3_^−^ alone cover the nitrogen requirements. Three series of experiments were run and the conditions tested in each are recalled in Table 1.

In experiment #1, a first set of four cultures was run under 12 h Light: 12 h Dark (LD) cycles. Two were started with elevated NO_3_^−^ concentrations (900 and 700 μmol NO_3_^−^ L^−1^) and two others with no N source in the growth medium. All four cultures were started from the same seed culture and grown in the same incubator, at the same incident irradiance of 240 µE m^−2^ s^−1^ for 12 h per day. NO_3_^−^ concentrations and cell counts were monitored in time. To verify if any decrease in nitrate concentration could be related to some other (possibly abiotic) process but uptake by cells, a test was also conducted in which two additional vessels, which only contained nitrate-rich medium (800 μmol NO_3_^−^ L^−1^) and no cells, were exposed to the same LD regime. NO_3_^−^ concentrations were monitored over time for several weeks. This control served as baseline for the NO_3_^−^-containing culture experiments.

In Experiment #2, two more culture vessels were started as in experiment #1 with elevated NO_3_^−^ concentrations (1038 μmol NO_3_^−^ L^−1^), but exposed to continuous light (LL). The incident irradiance was set to 120 µE m^−2^ s^−1^ for 24 h per day, so as to provide the cultures with the same total daily light dose as in the other experiments. NO_3_^−^ concentrations and cell counts were monitored over time.

Experiment #3 is an expansion of Experiment #1, in which we tested two nitrate levels (Experiment 3.1: 800 and Experiment 3.2: 90 μmol NO_3_^−^ L^−1^) and compared them to NO_3_^−^-free cultures run in parallel and taken as reference. Cultures were run in triplicate and exposed to a 12 h:12 h LD regime. Although the same setup was used, a careful measurement of the average incident irradiance field in the incubator gave 220 µE m^−2^ s^−1^ in Experiment 3.1 and 260 µE m^−2^ s^−1^ in Experiment 3.2. Rates of net N_2_ fixation and incorporation into the biomass were monitored using the stable isotope tracer method described by Montoya et al. [33], using 99% pure ^15^N_2_ gas (Eurisotop, Saint-Aubin, France). Incubations were performed in polycarbonate bottles. In Experiment 3.1, a volume of 0.5 mL ^15^N_2_ gas was injected in 170 mL bottles that contained a 0.7 mL residual air bubble, leading to an initial enrichment of 24%. In Experiment 3.2, due to volume restrictions, a volume of 0.1 mL ^15^N_2_ was injected in 20 mL bottles that did not show any residual air bubble, leading to an initial enrichment of 45%. Nitrogen incorporation per cell was then derived using the measured cells’ abundances. Incubations were started at the beginning of the dark phase and left for 24 h. Because cells divide (if they divided that day) at the mid light phase [31], i.e., after the period of N_2_ fixation, we logically referred to the cell abundance at the beginning of the incubation to estimate a nitrogen production per cell. Initial ^15^N abundances in the biomass were regularly verified at the light to dark transition and proved similar to the natural abundance, i.e., 0.366%. Following the incubation period, samples were filtered through pre-combusted (450 °C) Whatman GF/F filters (25 mm in diameter, nominal porosity ≈ 0.7 μm) using a low vacuum pressure (<100 mm Hg). Following filtration, filters were placed into 2 mL glass tubes, dried for 24 h in a 60 °C oven and stored dry until laboratory analysis. These filters were used to determine the final ^15^N enrichment ratio in the particulate organic matter on an Integra-CN mass spectrometer (SERCON) calibrated using glycine references according to Raimbault and Garcia [34]. Nitrogenase activity was recorded in Experiment 3.1.

## 3. Results

In this study, both nitrate and nitrite were monitored in the vessels. Nitrite concentrations always remained close to zero; therefore, we pooled NO_3_^−^ + NO_2_^−^, and considered this sum as being nitrate. The growth rates obtained in the different experiments are summarized in Table 1.

### 3.1. Control

In 9 days, the two control reactors devoid of biomass show a very similar trend with a steady 5.09% and 6.63% linear decrease in NO_3_^−^ concentrations. The corresponding slopes estimated over the whole series are −5.68 and −8.13 μmol NO_3_^−^ L^−1^ d^−1^, thus averaging −6.91 μmol NO_3_^−^ L^−1^ d^−1^. Bacterial contamination in these vessels remained negligible; therefore, some small, yet non negligible and consistent NO_3_^−^ disappearance occurred with time that was not due to *Crocosphaera* or bacteria uptake. In the following experiments, we operated a baseline correction by removing the average decreasing trend from measured NO_3_^−^ concentrations. Note that this correction remains minor and doesn’t change the main results or the conclusions developed thereafter.

### 3.2. Culture Experiments

#### 3.2.1. Experiment #1

Growth rates recorded under the LD regime in the NO_3_^−^-free culture medium are 0.43 (R^2^ = 0.98) and 0.35 (R^2^ = 0.99) d^−1^. Those observed in the two NO_3_^−^-rich cultures are 0.30 (R^2^ = 0.93) and 0.31 (R^2^ = 0.96) d^−1^. These growth rates do not statistically differ at the 5% level (Student t test). The total carbon and nitrogen contents per cell are needed to deduce the total nitrogen and carbon biomass formed in this experiment #1. We pooled data of carbon and nitrogen content in *C. watsonii* from the present study as well as our past studies [29,31,35] to derive an average content in our strain. The most robust reference appears to be the volumetric content as it allows comparison of N contents in cells with different sizes. We obtained an average of 25.62 ± 3.71 fmol C μm^−3^ and 2.69 ± 0.46 fmol N μm^−3^. The average cell volume in this Experiment #1 is 12.35 μm^3^. Therefore, the average N content is 33.22 fmol N cell^−1^. Given the number of cells formed over time in the two NO_3_^−^-rich cultures, the according total PON buildup at the end of the experiment is 679.3 and 796.3 μmol N L^−1^ (Figure 1a, closed and open black circles and the black continuous and dashed lines showing the according fitted dynamics). Based on the observed population dynamics, we also simulated what the expected changes in NO_3_^−^ concentrations should be, if DIN is to interrupt the activity of N_2_ fixation and *Crocosphaera* switches to NO_3_^−^ uptake to support its growth. The thick grey lines on Figure 1a predict how NO_3_^−^ concentrations should evolve in both culture replicates if NO_3_^−^ is taken up to support all of the observed PON buildup in the biomass over time. That is, if NO_3_^−^ is the primary source of N in the biomass, the PON buildup should be reflected in a symmetrical decrease in the NO_3_^−^ concentration: PON and NO_3_^−^ are plotted with the same scale for this purpose. Simulations thus predict that the initial NO_3_^−^ concentration would be drawn to zero within 10 and 9 days in the two replicates. In clear contrast, the actual NO_3_^−^ concentrations measured in the medium show a slightly decreasing trend with a slope of −7.35 and −1.89 μmol NO_3_^−^ d^−1^ in the two replicates (Figure 1a, closed and open grey diamonds). After 10 days, NO_3_^−^ concentration had only decreased by 7.72% and 2.52% in the two replicates. That is, in reality NO_3_^−^ uptake supported at most 10.4% of the biomass formed in the first culture replicate and 2.3 in the second. The hypothesis of a NO_3_^−^-supported growth (visualized by the grey lines on Figure 1a) does not match the observed evolution of NO_3_^−^ in the culture, and must therefore be refuted.

#### 3.2.2. Experiment #2

The growth rates obtained in the two (NO_3_^−^-rich) cultures exposed to continuous illumination are 0.19 (R^2^ = 0.99) and 0.22 (R^2^ = 0.99) d^−1^, which is about 50% slower than the growth rates observed under light:dark cycles for the same 24 h-integrated light dose. The average cell volume in this experiment is 18.06 μm^3^. Therefore, the average N content is 48.52 fmol N cell^−1^. The total amount of PON formed over the 15-day growth period is 411 and 570 μmol N L^−1^ (Figure 1b, closed and open black circles). As in Experiment 1, we simulated how NO_3_^−^ concentrations in the medium should evolve, if biomass formed in this experiment is supported by NO_3_^−^ uptake (Figure 1b, solid and dashed grey lines). Given the observed biomass dynamics, we predict that NO_3_^−^ concentrations after 15 days should have been drawn down by 60% and 43% of the initial concentration in the two culture replicates (Figure 1b, thick grey lines). However, the NO_3_^−^ concentration actually decreased by 9.7% and 18.2% (Figure 1b, closed and open grey diamonds). In the end, NO_3_^−^ uptake may have supported at most 24.55% and 27.54% of the observed biomass increase. Here again, the hypothesis of a NO_3_^−^ supported growth does not agree with the data. Despite the very unfavorable conditions that the continuous irradiance imposes on the nitrogenase, most of the nitrogen in the biomass is acquired through N_2_ fixation in the two LL cultures.

#### 3.2.3. Experiment #3

In this experiment, we monitored the population dynamics in triplicate cultures over several growth phases, i.e., in between repeated dilutions of the culture medium.

##### Experiment #3-1

NO_3_^−^-rich cultures were started with an initial concentration of 800 μmol NO_3_^−^ L^−1^. During exponential growth, we obtained a total of six recorded maximum growth rates per treatment. Their average is 0.32 ± 0.02 d^−1^ in the NO_3_^−^-rich cultures and 0.34 ± 0.02 d^−1^ in the NO_3_^−^-free cultures (Figure 2). An independent-samples Student *t*-test run on the two sets of six growth rates (*p* = 0.57) indicates that they do not statistically differ between the treatments. Therefore, we conclude that cells grew equivalently well whether cultures were devoid of, or supplied with, NO_3_^−^. Another *t*-test performed on the cell sizes (*n* = 91 for the NO_3_^−^-rich cultures and *n* = 68 for the NO_3_^−^-free cultures; *p* = 0.09) indicates that cell sizes do not statistically differ either between treatments.

The average N cellular content measured in the NO_3_^−^-rich cultures is 62.47 ± 9.74 fmol N cell^−1^. The total PON formed during the experiment is 930.6 and 1357.8 μmol N L^−1^ in replicates A and B, respectively, and 686.7 and 567.1 μmol N L^−1^ in the two growth phases of replicate C (Figure 2). The total amount of nitrate that disappeared from the cultures over the exact same period is null in the first two replicates, and 71.8 and 9.29 μmol NO_3_^−^ L^−1^ in the third replicate (Figure 2). In the end, NO_3_^−^ uptake supported none of the biomass formation in the first two replicates, and at most 10.5 and 1.6% of the biomass increase in the third replicate.

We performed a total of 18 ^15^N_2_ incubations on samples from the NO_3_^−^- rich cultures and 18 on samples from the NO_3_^−^- free cultures. The according N production estimated from the ^15^N_2_ incorporation is 1.37 ± 0.75 fmol N cell^−1^ d^−1^ in the NO_3_^−^-rich cultures and 1.99 ± 0.72 fmol N cell-1 d-1 in the NO_3_^−^-free cultures. The daily N production estimated by the isotopic method represents 2.59 ± 1.43% of the cellular N in the NO_3_^−^-rich cultures and 5.15 ± 2.11% of the cellular N in the NO_3_^−^-free cultures. Although the measured rates of ^15^N_2_ incorporations are very low in both treatments, they show the same order of magnitude, suggesting that similar N_2_ fixation activities occurred in the two treatments. An independent-samples Student *t*-test run on the two sets of N_2_ incorporation rates (*p* = 0.14), confirms that the biomass increase supported by N_2_ fixation is not significantly different between the two treatments. Therefore, we conclude by observing equivalent N_2_ incorporation rates between NO_3_^−^-free and NO_3_^−^-rich cultures.

Nitrogenase activity recorded using on-line GC incubations revealed very similar dynamics in the NO_3_^−^-rich cultures and the NO_3_^−^-free cultures. The average records taken on the NO_3_^−^-rich and NO_3_^−^-free cultures are shown on Figure 3. Nitrogenase activity becomes detectable after about 3 h into the dark phase, peaks around mid-dark, and decreases thereafter to arrive back at undetectable levels by the end of the dark. Some day-today variability can be observed but with very similar magnitude between the two treatments. Records were taken throughout the entire experiment period as cells were growing exponentially, indicating that no significant change in nitrogenase activity occurred in the course of the experiment. The average, gross amount of nitrogen fixed per day in this experiment is 496.7 ± 203.1 fmol N cell^−1^ d^−1^ (*n* = 6) in the NO_3_^−^-rich cultures and 510.4 ± 186.2 fmol N cell^−1^ d^−1^ (*n* = 4) in the NO_3_^−^-free cultures. As already reported previously [4,31,35,36,37], these gross rates derived from the enzyme activity largely exceed the net growth requirements. The ratio between the gross amount fixed to the total nitrogen content in cells spans from 4.8 to 13.8 in the NO_3_^−^-rich cultures and from 5.2 to 14.0 in the NO_3_^−^-free cultures. Given the respective growth rates of 0.32 and 0.34 d^−1^ in the NO_3_^−^-rich and the NO_3_^−^-free cultures, each cell needs a net gain of 17.9 and 17.7 fmol N per day on average, meaning that the estimated gross enzyme activity was on average 27.7 and 28.8 times larger than the net N requirements in the NO_3_^−^-rich and the NO_3_^−^-free cultures, respectively. Overall, the recorded enzymatic activity of the nitrogenase is thus extremely similar between the treatments, so totally unaffected by the presence of nitrate in the cultures.

##### Experiment #3-2

In this experiment, the NO_3_^−^-rich cultures were started with an initial concentration of 90 μmol NO_3_^−^ L^−1^. The population dynamics monitored in the triplicate cultures shows an average, exponential growth rate of 0.39 ± 0.03 d^−1^ in the NO_3_^−^-rich cultures and 0.37 ± 0.06 d^−1^ in the NO_3_^−^-free cultures. Again, these exponential growth rates are not statistically different (independent-samples Student *t*-test; *p* = 0.57) whether cultures are supplied with NO_3_^−^ or not. Nitrogen content in the biomass averages 55.44 ± 1.44 fmol N cell^−1^ in the NO_3_^−^-rich cultures. PON samples were acquired from day 9 to 16. We used this measured, average PON content per cell and the population dynamics measured in numbers over the first nine days to reconstruct the dynamics expressed in mass of nitrogen in the NO_3_^−^-rich cultures (Figure 4, black lines). Over the first nine days, the PON in the biomass increased by 220.8 ± 22.5 μmol N L^−1^. The hypothetical, simulated NO_3_^−^ consumption (grey line on Figure 4), should cell growth be supported by NO_3_^−^ uptake, indicates that the NO_3_^−^ stock should have been depleted in 6 days. Instead, NO_3_^−^ concentrations remained essentially constant in the three replicates over the same period: the deviation around the mean concentration is 0.94, 1.27 and 2.09 μmol N L^−1^). Between days 8 and 16, the biomass increased by 584.8 ± 81.7 μmol N L^−1^ in the three NO_3_^−^-rich cultures. The simulated NO_3_^−^ consumption during the second phase of growth (from day 8) predicts a dramatic decrease in NO_3_^−^ concentrations, due to the already higher level of biomass compared with that in the beginning of the experiment. There, NO_3_^−^ should in theory be depleted in less than two days to cover for the PON buildup in the biomass (Figure 4). Instead, the measured NO_3_^−^ concentrations decreased by 12.8 μmol L^−1^. In the end, over this 15-day experiment, the average biomass formed is 777.53 ± 58.84 μmol N L^−1^, while only 14.6 ± 2.66 μmol NO_3_^−^ L^−1^ disappeared from the cultures. NO_3_^−^ uptake supported at most 1.90 ± 0.45% of the nitrogen biomass formed in the three NO_3_^−^-rich cultures.

## 4. Discussion

Biological nitrogen fixation in the open ocean represents a critical entry point for new nitrogen into the marine N cycle. Yet the accuracy of current estimations of global N_2_ fixation rates remains an open debate, with evidence pointing to their systematic underestimation. One long standing, questionable paradigm suggests that open ocean diazotrophs will not be found fixing nitrogen in areas where NO_3_^−^ is available. In the present analysis, we prove that nitrogen fixation rates in the unicellular cyanobacterium *C. watsonii* WH8501 are not hindered in the presence of NO_3_^−^ when light energy is sufficient to support nitrogen fixation.

The first main message that we derive from these experiments is that, in all the experiments we performed under a natural light regime (i.e., under light:dark cycles), NO_3_^−^ provision did not stimulate the growth rates, nor did the high NO_3_^−^ concentrations show any deleterious effect on growth efficiency. The fact that the exponential growth rates, derived from the counts over repeated experiments, are not significantly different between treatments is irrefutable proof that overall growth, in terms of population dynamics, was not enhanced in the presence of nitrate.

As cell sizes were also similar in NO_3_^−^-rich and NO_3_^−^-free cultures, the net incorporation of N (whatever its origin) and C is expected to be equivalent, whether in terms of rate or stock in the cells. It could be that cultures incorporated equivalent amounts of N in both treatments, but that cultures in the presence of NO_3_^−^ switched from N_2_ fixation to NO_3_^−^ uptake. From their analysis, Garcia and Hutchins [26] deduced that available light energy drives the growth rate, which results in an according nitrogen demand; cells take up DIN when available and, if need be, N_2_ fixation provides the complement to meet this N demand. If that is also the case in our experiments, we should observe a positive incorporation of NO_3_^−^ in cells as well as an exponential decrease in NO_3_^−^ concentration in the cultures, matching the exponential growth of the biomass, since these cultures are closed systems. The amount of nitrate disappearing from the culture following this uptake tells us to what extent NO_3_^−^ use supports cell growth. The observation of the NO_3_^−^ concentrations in all experiments indicate that the nitrate possibly taken up by *Crocosphaera* only supported a small percentage of the biomass produced (Figure 1, Figure 2 and Figure 4). Note that, had the baseline correction not been applied, results would have changed by a couple of percentage points only, which does not affect the main results discussed here, i.e., a striking absence of significant use of NO_3_^−^ for growth. Besides, the changes in NO_3_^−^ concentrations were so small that linear fits gave more robust results than exponential decreases. Yet, if NO_3_^−^ uptake by the biomass is driven by the exponential growth of the population, the decreasing trends in NO_3_^−^ concentration should be exponential: they never were in our experiments, no matter the NO_3_^−^ availability. Note that Garcia and Hutchins [26] performed their experiments on *Crocosphaera watsonii* strain WH0003, not WH8501 as in the present study. A major difference thus appears between these two strains, in the fact that DIN is the primary N source for WH0003 while WH8501 does not seem to make use of NO_3_^−^ to support their growth and almost completely relies on N_2_ fixation.

Logically, if cells are to switch to NO_3_^−^ use, the N_2_ fixation rate is also expected to decrease accordingly. At the irradiance level applied in our study, we observed no statistically significant reduction in N_2_ fixation in the cultures grown in nitrate. Instead, as shown by ^15^N_2_ incubations, cells cultivated with a non-limiting supply of nitrate keep fixing N_2_ at equivalent rates as cells maintained under obligate diazotrophy. The specific, nitrogen production derived from nitrogen fixation, both per cell or per um^3^ (fmol N cell^−1^ d^−1^ or fmol N μm^−3^ d^−1^), is equivalent in all cultures, whether provided with NO_3_^−^ or not. Very consistent with these results are the on-line records of nitrogenase activity (Figure 3), which reveal an unaffected daily dynamics of the enzymatic activity in the NO_3_^−^-rich cultures compared to that in NO_3_^−^-free cultures, as well as an efficient activity leading to equivalent daily gross estimates of fixed N_2_. Note that we express some reservations about the estimated rates of N_2_ incorporation derived from the isotopic analyses as their magnitude is far lower than the N incorporation deduced from PON quantifications. As argued by Grosskopf et al. [38], the ^15^N method tends to produce systematic underestimations of N_2_ fixation rates. If we do not rule out such possibility, we at least strongly believe that the actual enrichment remained consistent throughout the incubations, making our results valid in terms of proportion between treatments.

Another reason why *Crocosphaera* did not start to use NO_3_^−^ might be that, due to the acclimation phase, cellular reserves were replenished, which would have made them less dependent upon external sources. Although we are not sure whether and how many times a cell may divide on its sole reserves, it is reasonable to think they cannot store enough N to sustain many divisions because N is the second major component of the biomass and cells are sterically limited. When cells grow at 0.3 to 0.4 d^−1^, they divide every ~1.7 to 2.3 days. So even if one or two divisions were supported by internal reserves, cells would need to replenish these again after a few days and this demand should then show on the NO_3_^−^ concentration data. No such lag phase in NO_3_^−^ consumption appears in any of the experiments, whose duration is very long compared to the division time. Whether cells are provided with nitrate or not, their reserves are replenished by N_2_ fixation. We thus can rule out any possible “internal reserve effect” on NO_3_^−^ uptake.

*Crocosphaera* not using NO_3_^−^ could be explained by the expression of nitrate assimilation genes being cryptic in this strain. However, in a global transcriptomic profiling of gene expression of this strain grown under N_2_, the transcripts for the *narB* gene encoding the nitrate reductase, the *nirA* gene involved in nitrate and nitrite reduction as well as the gene *nrtP* encoding high-affinity nitrate/nitrite bispecific transporter have been detected [39]. A specific analysis of the transcription of these genes under the experimental conditions used in our study will help in confirming these data. If these genes are actually expressed, it would be relevant to uncover the post-transcriptional regulation that could explain why the strain does not use nitrate. We sought to restrain N_2_ fixation by placing cells in conditions very unfavorable to nitrogenase activity. Nitrogenase being irreversibly inactivated by oxygen [40], unicellular, diazotrophic cyanobacteria has evolved a temporal separation of nitrogen fixation and of the oxygen-evolving photosynthesis, by confining nitrogenase synthesis and activity to the dark hours. The cultures exposed to continuous light did survive, but they showed here about a 33% reduction in their growth rate, compared to the cultures exposed to a L:D regime with the same daily light dose. This reduced growth rate could either be due to a lower, daily integrated photosynthetic efficiency related to the lower, instantaneous irradiance in the LL cultures, or to a negative impact of photosynthetically-evolved O_2_ on the nitrogenase enzyme, or both. In any case, this lower growth rate also implies a slower rate of nitrogen incorporation. The higher disappearance of nitrate from the culture medium compared to that under the LD regimes (Figure 1) suggests that the stress imposed by the continuous illumination did provoke a consumption of NO_3_^−^ by *C. watsonii*. Therefore, the nitrate uptake pathway seems functional in this strain. Yet the predicted NO_3_^−^ consumption required to fully support this growth still largely deviates from the actually observed NO_3_^−^ concentrations (Figure 1B). Despite the stress imposed on the nitrogenase and the non-limiting availability of NO_3_^−^, N_2_ fixation supported more than 75% of the biomass produced. We therefore conclude that in *C. watsonii* strain WH8501, N_2_ fixation is the primary pathway for N acquisition while NO_3_^−^ is hardly used at all under natural (i.e., LD) light cycles. One may wonder whether, under very low irradiances, the additional costs imposed by N_2_ fixation [2,3] would not force cells to switch to NO_3_^−^ uptake. Although this remains to be verified, we strongly suspect that cells will keep fixing N_2_ because they did not mainly switch to NO_3_^−^ uptake under a continuous light despite the severe stress that this imposes on N_2_ fixation.

The present results demonstrate the existence of an NO_3_^−^-insensitive cyanobacterial diazotrophy adding to a growing body of literature which modifies long-persisting paradigms of marine diazotrophy. In particular, we wish to encourage reconsideration of the N_2_ fixation process in biogeochemical models and the contribution of unicellular cyanobacteria to oceanic nitrogen fixation rates. N_2_ fixation is now explicitly included in biogeochemical models [41,42,43] but there are still significant issues with the placement of diazotrophs in areas rich in fixed nitrogen (relative to phosphorus), such as the North Atlantic [18,19]. This is related to the fact that the environmental controls on N_2_ fixation are still partially understood and insufficiently constrained in global models. Another difficulty is the still sparse availability of in situ data of abundance and activity of diazotrophs, which does not make model calibration any easier. Based on the hypothesis that N_2_ fixation should occur, or become competitive, when nitrate is limited, the process can be prescribed as a function of nutrients availability [44], assuming that the global N budget is balanced. The models HAMOCC [45,46] and MOPS [47] for instance, impose a constant biomass of diazotrophs and follow the above-mentioned concept by adjusting the rate of N_2_ fixation so that it balances the deviations of the NO_3_^−^:PO_4_^3−^ ratio away from the Redfield [48,49] canonical value. Most parameterizations of N_2_ fixation are restricted to regions with observed low NO_3_^−^:PO_4_^3−^ ratios, where it is prescribed to top up denitrification losses [46].

As emphasized by Kriest and Oschlies [47], predicting oceanic N fluxes is far from trivial, as they occupy a wide range of spatio-temporal scales that span from metabolic reactions within micrometer-large cells to the global circulation. Furthermore, the process of N_2_ fixation is all the more complex to describe, as it is operated by organisms with a variety of physiologies. Historically, the filamentous cyanobacterium *Trichodesmium* spp. was thought to be the major diazotroph in the open ocean [10], until the generalization of molecular tools revealed diverse diazotrophic capabilities in unicellular cyanobacteria (thereafter named UCYN) [9,22,23] as well as in heterotrophic bacteria [8,50]. We believe that while tackling the incongruence of nitrogen budgets at large scales, some answers will be found when looking at how N_2_ fixation operates at small scales, allowing for new parameterizations of the process to emerge in global models. Our results clearly suggest that, assuming that similarly behaving diazotrophs can also be found in nature, the occurrence of diazotrophs in the world ocean should not be delineated by availability of NO_3_^−^. While diazotrophs may still be distributed by competition for nutrients, it may be that NO_3_^−^ is less of a factor than was once considered. This may suggest that modelled, global rates of N_2_ fixation are underestimated due to the reduced overall biomass predicted in areas of significant NO_3_^−^ concentration. Along with new results which have revealed a wider phylogeny of diazotrophs and their occurrence in areas where they were not suspected to occur, such as the polar regions (e.g., [51,52,53]), and the deep ocean (e.g., [54,55]), our results suggest that the study of marine diazotrophy is in need of expansion. This may help to alleviate the debate regarding global underestimation of marine N_2_ fixation which has been ongoing for the last two decades.

## Figures and Tables

**Figure 1 microorganisms-09-02073-f001:**
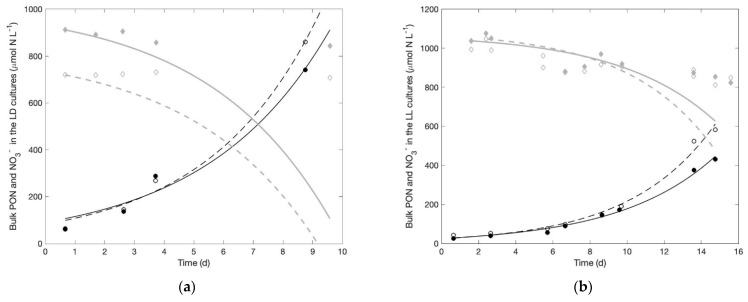
Measured and fitted population dynamics of *Crocosphaera* expressed as particulate organic nitrogen (PON, μmol N L^−1^) and NO_3_^−^ concentrations (**a**) in the two LD, NO_3_^−^-rich culture replicates from Experiment #1 and (**b**) in the two LL, NO_3_^−^-rich culture replicates from Experiment #2. Measured biomass concentration (closed and open black circles) (μmol N L^−1^) and the corresponding, measured nitrate concentrations (closed and open grey diamonds) in the cultures. Black lines (continuous and dashed) represent the fitted cell biomass and thick grey lines (continuous and dashed) the predicted changes in NO_3_^−^ concentration that would result from a growth entirely supported by NO_3_^−^ uptake.

**Figure 2 microorganisms-09-02073-f002:**
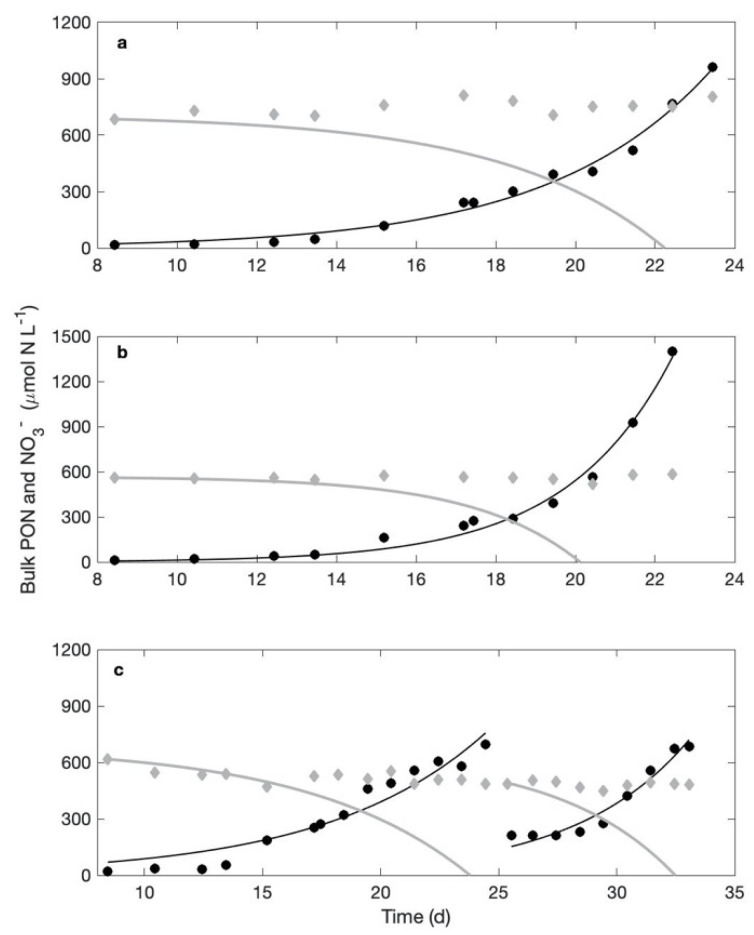
Experiment 3-1. Measured (closed, black circles) and fitted (black line) biomass dynamics expressed as particulate organic nitrogen (PON, μmol N L^−1^) and measured NO_3_^−^ concentration (grey, closed diamonds) in the three culture replicates (**a**–**c**) of the NO_3_^−^-rich cultures. The grey line is the predicted changes in NO_3_^−^ concentration that would be observed should nitrate uptake support 100% of the biomass formed during the experiment.

**Figure 3 microorganisms-09-02073-f003:**
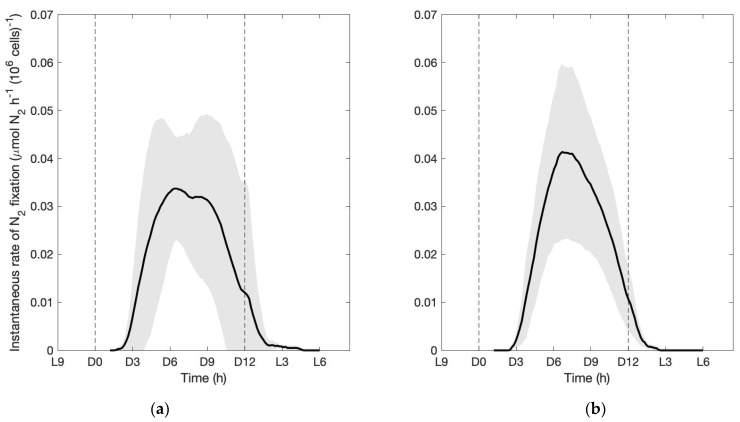
Experiment 3-1. Nitrogenase activity in (**a**) the NO_3_^−^-rich cultures and (**b**) the NO_3_^−^-free cultures. The average, real-time N_2_ fixation rate recorded on the cultures (black line), expressed in μmol N_2_ fixed per hour and per million cells, is shown with its standard deviation (grey shaded area). Vertical dashed lines indicate the onset of the dark (at D0) and onset of the light (after 12 h of dark, at D12 = L0).

**Figure 4 microorganisms-09-02073-f004:**
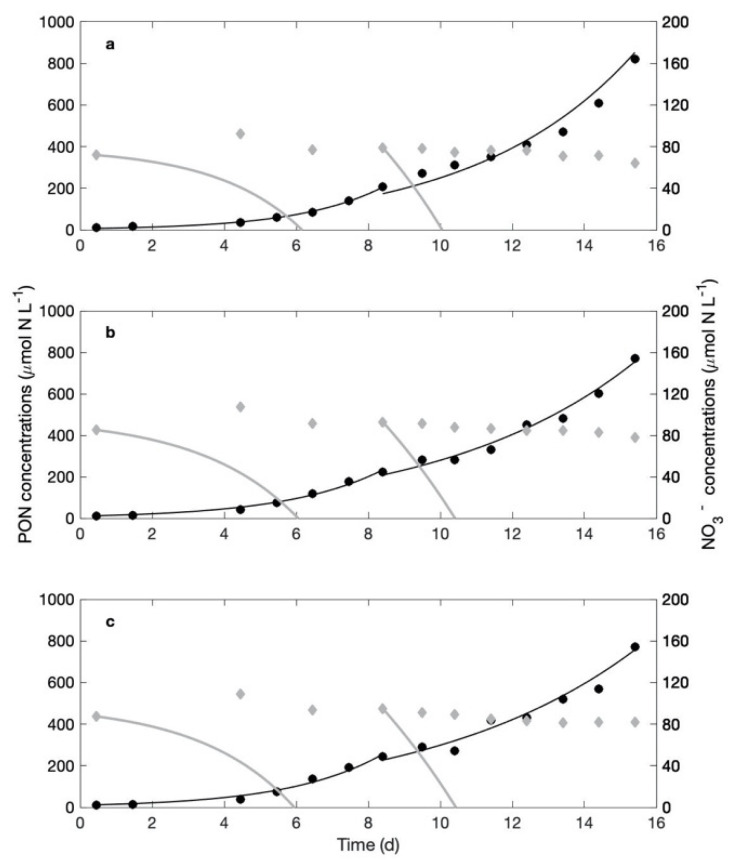
Experiment 3-2. Measured (closed, black circles) and fitted (black line) biomass dynamics expressed as particulate organic nitrogen (PON, μmol N L^−1^; left axis) and measured NO_3_^−^ concentration (grey, closed diamonds; μmol NO_3_^−^ L^−1^; right axis) for two successive growth sequences in the three culture replicates (**a**–**c**) of the NO_3_^−^-rich cultures. The grey lines are the predicted changes in NO_3_^−^ concentration that would be observed should nitrate uptake support 100% of the biomass formed during the experiment. PON values are measured directly from day 9 to 16 and converted from cell counts during the first 8 days; a conservative approach was to split the analysis of this experiment into two phases (days (0–8) and (9–16)). Note the difference in scale between the two Y axes.

**Table 1 microorganisms-09-02073-t001:** Summary of the experimental conditions tested, and data acquired, in the three series of experiments. LD, Light:Dark. LL, Light:Light (i.e., continuous light). (+) and (−) indicate that cultures were grown with or without NO_3_^−^. Max NO_3_^−^ support, maximum percentage of the N incorporated in the biomass possibly supported by NO_3_^−^ uptake. The symbol ^∅^ indicates that averages measured in the NO_3_^−^-rich and NO_3_^−^-free cultures (in a given experiment) are not statistically significant at the 5% level.

Condition	Experiment #1		Experiment #2	Experiment #3			
				3.1		3.2	
Light regime	LD		LL	LD		LD	
irradiance (µE m^−2^ s^−1^)	240		120	220		260	
NO_3_^−^ (μmol N·L^−1^)	(+)	(−)	(+)	(+)	(−)	(+)	(−)
Growth rate (d^−1^)	0.30, 0.31	0.43, 0.35 ^∅^	0.19, 0.22	0.32 ± 0.02	0.34 ± 0.02 ^∅^	0.39 ± 0.03	0.37 ± 0.06 ^∅^
N content (fmolN cell^−1^)	33.22		48.52	62.47 ± 9.74		55.4 ± 1.44	
Max NO_3_^−^ support (%)	10.4, 2.3	NA	24.55, 27.54	0, 10.5, 1.6		1.90 ± 0.45	

## Data Availability

All the data are shown in the manuscript and can be obtained from the authors upon request.

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
