# Peer review of "Independence of a Marine Unicellular Diazotroph to the Presence of NO3"

_microorganisms, 2021, doi:10.3390/microorganisms9102073_

Round 1

Reviewer 1 Report

The work is relevant to the field and has the potential to provide important data and conclusions. However, there is a tendency to repeat the results in the discussion.

An explanation of the predicted calculations should be provided in the materials and methods.

Does the authors have relevant information on nitrogenase fixation regulation and NO3- uptake regulation?

Minor comments:

Lines 50, 76, 456: in situ should be in italic

Line 78: N2, 2 should be in subscript

Line 145: remove space between 27 and °C

Figure legends: NO3, 3 should be in subscript

Reviewer 2 Report

The authors state that some unicellular diazotroph cyanobacteria are independent of the presence of NO3, but instead keep fixing N2. Through a set of experiments, they prove their statement, modifying long-persisting paradigms of marine diazotrophy. Thus, although the data are very preliminary, this is a very interesting paper, which brings a new approach to N2 fixing marine microorganisms’ ecology and the N2 fixation process in biogeochemical models.

The discussion is well reasoned and conservative in the sense that many other factors must be considered in the ocean biogeochemical nitrogen cycle, but the data are sufficiently robust to raise questions as to if the occurrence of diazotrophs in the world ocean should be or not delineated by NO3 availability. The data presented suggest that they shouldn’t.

The introduction contains information relevant to the subject researched, the objective, and the importance of the study are well defined and highlighted.

The experimental design is adequate but could be clearer.

Results are well presented and are easily perceptible, although some minor changes can be made to improve the clarity.

The analysis of the scientific literature is comprehensive, the authors have consulted numerous relevant and current articles.

Therefore, I recommend the publication of the paper with minor changes.

Detailed recommendations:

Material and methods:
Line 117 – what exactly is the sufficient frequency used?

Line 125 – what do you consider to be “sufficient growth”? if you only started measuring metabolic parameters after this “sufficient growth” how did you determine it was time to start the measurements?

Line 134 – I believe the manufacturer is Perkin Elmer

Line 158 – In experiment #2 you changed 2 parameters photoperiod and NO3- concentration)? Two replicas.

Line 163 – So, in experiment #3, you changed nitrate levels, light irradiance, flask volume, and N2 incorporation? Four parameters? Three replicas.

I advise you to make a table with all the variables, to make it easier for the reader to understand the experiments.

Results:

Line 231 – state what the dashed line stands for in the text and the caption of figure 1.

Figure 1a – why are there so much missing data for figure 1a, between days 4 and 9? Measurements should be much more systematic.  You base your conclusions on one final measurement, which is weak if I’m reading well.

Line 244 – Are there any statistical differences between treatments?

Line 251 – solid and dashed grey lines?

Figure 1b – more consistent data, for 2 and 3 measurements were performed.

Line 258 – There is no need to repeat the experimental design

Line 271 – Again, a table with these results would help the reader to understand the N content, PON content, growth rate, etc. between all treatments. And if there are any statistical differences.

Figure 2 – this could be only one graphic with medium and standard deviation. It would give more information to the reader. Why did you do another batch for replica 3? This is not explained in the text, and I think it doesn’t make much sense. SD would be very large for the three replicas, mostly for PON.

Figure 4 – again, this could be only one graphic with medium and standard deviation.

Discussion:
line 446 - You can’t dissociate the continuous light discussion from the possibility of occurring photo-oxidation and carbon fixation from photosynthesis, besides stress caused to nitrogenase. Lower growth rates due to the lower carbon fixation also require less nitrogen uptake.

Reviewer 3 Report

The authors presented results of three experiments of growth with the diazotroph Chrocosphaera in order to test the hypothesis of nitrate availability in the medium decrease the N2 fixation. The results indicate that the main source of N for growth of the strain of Chrocosphaera used was N2 fixed. Based on these findings, the authors propose re-considerer the paradigm that the N2 fixation is constrained to oceanic areas limited by DIN. The manuscript is well written and the results are highly interesting. However, I have several global comments:

-The interaction between DIN concentration and growth depends on both DIN availability and internal N pool that in turn depends on the previous story of the cell.  The experiments were performed with cells acclimated to unlimited DIN conditions, possibly with full internal N pool which would explain low rates of nitrate uptake. I do not know the physiology of this species and I am confused regarding to its capacity of N storing which would be mobilized when the growth conditions change. I miss in the manuscript a discussion more based on description of the N metabolism  of this species.  

-Nitrate usage is calculated from its removal in the culture medium  that is an approach fairly crude.  However, N2 fixation is calculated by means of incubations in 15N2 or nitrogenase activity. I am not sure that all these measurements can be directly (and quantitatively) compared. To me, taking into account the objective of the manuscript, nitrate upkate rates also should have being measured with experiments of uptake of 15N-NO3-. Otherwise, the calculations about the amount of growth depending on nitrate and N2 fixation are uncertain. Additionally, see my below comments about the possible variation in cell N content.    

I have also several additional comments that I expect the authors find useful:

-Line 19. N2 fixation was only determined in one of the experiments.

-Line 20. I could not see data of carbon fixation rate in the cultures. Stoichiometry data (i.e. C and N content) are not presented either.

-Line 42. Please, to define "significant concentrations" in quantitative terms.

-Line 99. I think that these concentrations are fairly high compared to the concentrations normally found in the open sea (and probably much higher than saturation concentration). I will expect that the authors indicate which realistic oceanic conditions are being simulated in their  experiments as they try to extrapolate their results to the natural ambient.

-Line 107. I wonder if the authors mean that cell density in the cultures kept enough low to avoid self-shading produced by the own cells. Please, to explain.

-Line 109. The concentrations of other macronutrients important for the growth (i.e phosphate) should be indicated as well as salinity and pH.

-Line 127. I could not see data of carbon fixation rates.

-Line 132. Please, to indicate which was the mean bacterial abundance in the cultures as well as which was the initial cell abundance of Chrocosphaera.

-Line 139. I guess that ammonium concentration was not measured in the cultures. I wonder if it is due to the authors assumed that this form of DIN was not present in their cultures.

-Line 147. I wonder which the replication level of the treatments was in the experiments #1 and #2. Only one replication would not be enough to guarantee statistical significant differences among treatments.

-Lines 158-164. Please, to indicate which were the controls for the experiment #2 and #3 (were they cultures free-nitrate?).

 -Lines 167-171. It is unclear in which phase of the treatments was measured  the N2 fixation

-Lines 192-199. Some explanation for this reduction in nitrate concentration that is not due to biological activity should be supplied. Note that if this decreasing rates are similar in all treatments, it would modify the nitrate concentration substantially in the treatment starting with 90 microM.

-Line 203. The procedure used to estimate growth rates should be explained in Material and Methods as well as the statistical tests used (it is unclear if Student t test is suitable for these comparisons taking into account that the replication level of the treatments was low). Additionally, the meaning of the R2 shown into brackets is not explained.

Line 214. I cannot understand why the authors calculated the PON at the end of the experiment from cell abundance and bio-volume if POC and PON was measured in the culture unless that these measurements were performed punctually.  If this is the case, the authors should demonstrate (or justify) that  the treatments did not affect the cell stoichiometry. There are abundant literature showing that N content per cell varies substantially under nitrate concentration changing and/or during different phases of the growth curve.

-Line 234-235. These figures are confused. It is unclear what the continuous and dotted line represent. Furthermore, the nitrate concentration actually measured in the cultures should be shown. If the points are the mean calculated from several replications, some deviation measurements should be shown. I also wonder how these fitting lines were calculated.

-Line 268. I wonder that these sizes of samples represent ('n'; are they different individual cells measured for each treatment at different days?).

-Line 286. "the two sets of N2 incorporation rates". Please, to indicate how many measurements were performed.

-Line 289. This control treatment (nitrate-free) for Experiment #3 is not mentioned in Material and Methods.

-Line 304. Perhaps it would be appropriate to show the time course of nitrogenase activity.

-Lines 373-374. I suggest that the author consider that changes in N cell content could be produced.

-Lines 394-396. I wonder which results would be most appropriate to be extrapolated to the ocean.

Reviewer 4 Report

Your work is excellent. In your figures I think it is better to enlarge circles and triangles to facilitate instant grasping of their information. A lot of people is going to look at them.

Round 2

Reviewer 3 Report

I thank the authors for responding to my comments. On overall, I am satisfied with the reply of the authors and the corresponding modifications that they have done in the text. Specifically, I think that my doubts about the level of replication of the treatments and physiology of the species have been resolved.

I think that the manuscript is useful for publication in its present form.